# T_1_-Positive Mn^2+^-Doped Multi-Stimuli Responsive poly(L-DOPA) Nanoparticles for Photothermal and Photodynamic Combination Cancer Therapy

**DOI:** 10.3390/biomedicines8100417

**Published:** 2020-10-14

**Authors:** Sumin Kang, Rengarajan Baskaran, Busra Ozlu, Enkhzaya Davaa, Jung Joo Kim, Bong Sup Shim, Su-Geun Yang

**Affiliations:** 1Department of Chemical Engineering, Inha University, 100 Inha-ro, Michuhol-gu, Incheon 22212, Korea; ejddmini@naver.com (S.K.); busraozlu17@gmail.com (B.O.); 2Department of Biomedical Science, Inha University College of Medicine, 366 Seohae-Daero, Jung-gu, Incheon 22332, Korea; baskrajan@gmail.com (R.B.); zayatuya@gmail.com (E.D.); jungjookim325@gmail.com (J.J.K.); 3Program in Biomedical Science & Engineering, Inha University Graduate School, 100 Inha-ro, Michuhol-gu, Incheon 22212, Korea; 4Inha Institute of Aerospace Medicine, Inha University College of Medicine, 366 Seohae-Daero, Jung-gu, Incheon 22332, Korea

**Keywords:** melanin-like poly(L-DOPA) nanoparticles, L-DOPA, photodynamic therapy, photothermal therapy, T_1_-positive MRI nanoparticles

## Abstract

In this study, we designed near-infrared (NIR)-responsive Mn^2+^-doped melanin-like poly(L-DOPA) nanoparticles (MNPs), which act as multifunctional nano-platforms for cancer therapy. MNPs, exhibited favorable π-π stacking, drug loading, dual stimuli (NIR and glutathione) responsive drug release, photothermal and photodynamic therapeutic activities, and T_1_-positive contrast for magnetic resonance imaging (MRI). First, MNPs were fabricated via KMnO_4_ oxidation, where the embedded Mn^2+^ acted as a T_1_-weighted contrast agent. MNPs were then modified using a photosensitizer, Pheophorbide A, via a reducible disulfide linker for glutathione-responsive intracellular release, and then loaded with doxorubicin through π-π stacking and hydrogen bonding. The therapeutic potential of MNPs was further explored via targeted design. MNPs were conjugated with folic acid (FA) and loaded with SN38, thereby demonstrating their ability to bind to different anti-cancer drugs and their potential as a versatile platform, integrating targeted cancer therapy and MRI-guided photothermal and chemotherapeutic therapy. The multimodal therapeutic functions of MNPs were investigated in terms of T_1_-MR contrast phantom study, photothermal and photodynamic activity, stimuli-responsive drug release, enhanced cellular uptake, and in vivo tumor ablation studies.

## 1. Introduction

Cancer is the second leading cause of death worldwide; in 2018, approximately 9.6 million people died from cancer [1]. Generally, cancer therapy begins with surgery followed by radiation and chemotherapy, and this combination protocol has been accepted as a gold standard. Surgery and radiation therapy allow precise eradication of locally occurring cancer nodules, while chemotherapy treats cancer cells that have spread to distant sites. However, this traditional regimen appears to have reached a therapeutic plateau. Some oncologists have raised concerns that this sequential combination protocol may precipitate the neo-formation of metastatic foci, and promote the growth of metastatic cancer cells during the postoperative period [2]. Zhan et al. performed a meta-analysis using the reported literature and identified that delayed adjuvant chemotherapy after surgery resulted in poorer survival of breast cancer patients [3].

Recently developed multifunctional nano-systems, with an all-in-one design that carries several therapeutic modalities in one system, now open up new areas of cancer treatment [4]. Multifunctional nanoparticles exhibit various therapeutic functions, such as fluorescence-based cancer imaging, photodynamic action, photothermal effects, and target-specific drug release [5]. Clinicians may inject these nanoparticles prior to surgery and utilize their versatile therapeutic functions during surgery, such as imaging-based surgical guidance, photodynamic therapy (PDT), photothermal tumor–tissue ablation, and triggered anti-cancer drug release. Certain multifunctional nanoparticles have shown dramatic therapeutic efficacy in preclinical animal studies, and are currently undergoing human clinical trials [6,7]. 

With respect to multifunctional nanoparticles, PDT, photothermal therapy (PTT), and chemotherapy may be the best combination of regimens as a result of their synergistic therapeutic effects [4,8]. In PDT, reactions between a photosensitizer and oxygen existing in tissue generate reactive oxygen species (ROS), which kill cancer cells by damaging cellular macromolecules in response to near-infrared (NIR) irradiation [4,7]. PDT can minimize side effects because clinicians can precisely control the area of NIR-irradiation and ROS generation in tumors instead of normal tissue [9]. In PTT, photo-absorbing materials, such as gold and carbon nanomaterials, generate heat which kills cancer cells. The NIR-generated thermal energy synergistically enhances the cellular uptake of nanomaterials, and triggers the release of drugs into cancer tissues [10,11,12]. Therefore, trimodal PDT/PTT/chemotherapy is a robust treatment compared to monotherapy as its components exhibit synergistic effects. 

Recently, melanin, which is a dark pigment naturally existing in living organisms, is considered as an attractive drug delivery agent because of its good biocompatibility, photo-absorbance, ability to bind drugs, and chelating metal ions [13,14]. Recent studies also showed that melanin-like polydopamine nanoparticles (PDA NPs) could be used as photothermal agents due to their ability to convert NIR light into heat and kill cancer cells [14].

In this study, we utilized melanin-like poly(L-DOPA) nanoparticles (MNPs) to achieve a multifunctional regimen of photothermal, photodynamic, chemotherapeutic, and targeted cancer therapy (Scheme 1). MNPs were fabricated via KMnO_4_-oxidative polymerization of L-3,4-dihydroxyphenylalanine (L-DOPA), in which the embedded Mn^2+^ functioned as a T_1_-weighted contrast agent for MRI. MNPs were further decorated with specific functional modalities, a photosensitizer (pheophorbide a; PheoA), anti-cancer drugs (doxorubicin or SN38), and cancer-targeting folic acid (FA). Dual laser (670 and 808 nm)-responsive therapeutic functions of MNPs were investigated in terms of photothermal and photodynamic activities, thermally-triggered drug release, target-specific and thermal-boosting cellular uptake, and T_1_-weighted MRI. The therapeutic efficacy of MNPs was evaluated in vitro using cancer cells, and in in vivo cancer-xenograft animal models.

## 2. Experimental Section

### 2.1. Materials

L-3,4-dihydroxyphenylalanine (L-DOPA), potassium permanganate (KMnO_4_), 1-ethyl-3-(-3-dimethylaminopropyl) carbodiimide hydrochloride (EDC), N-hydroxysuccinimide (NHS), cystamine (CYS), hexamethylenediamine (HMDA), pheophorbide A (PheoA), FA, doxorubicin hydrochloride (Doxo), 7-ethyl-10-hydroxycamptothecin (SN38), 9,1-dimethylanthracene (DMA), DL-dithiothreitol (DTT), and 2′,7′-dichlorodihydrofluorescein diacetate (DCFH-DA) were obtained from Sigma-Aldrich (St Louis, MO, USA). EZ-Cytox (water soluble tetrazolium assay) was purchased from DoGenBio (Seoul, South Korea). Ultra-pure water (18 MΩ) was used throughout the experiments.

### 2.2. KMnO4-Oxidative Synthesis of COOH End-Capped MNPs Using L-DOPA

Carboxyl acid end-capped MNPs were fabricated via KMnO_4_ oxidation of L-DOPA. L-DOPA (200 mg) was first dissolved in 190 mL of deionized water under vigorous stirring. After dissolving, 10 mL of KMnO_4_ solution (10 mM) was added to the L-DOPA aqueous solution and stirred at 50 °C for 12 h. The mixed solution was centrifuged at 15,000 rpm for 10 min and washed three times with deionized water. The obtained MNPs were dispersed in deionized water for further studies.

To further investigate the potential of MNPs as a multifunctional platform with tumor-targeting ability, FA was conjugated to MNPs (Folate-MNPs; FA-MNPs) to recognize folate receptors overexpressed on the surface of tumor cells, thereby preventing nonspecific uptake by normal cells (Appendix A).

### 2.3. Cellular Redox System-Activatable Photodynamic Design of MNPs

First, MNPs (20 mg) were dispersed in 10 mL of 0.1 M sodium acetate buffer, and then reacted with EDC (9.4 mg) and NHS (6.9 mg). After stirring for 1 h, cystamine (13.52 mg) was added to the mixture, and the reaction was allowed to proceed for 12 h at room temperature. The cystamine-conjugated MNPs were retrieved by repeated centrifugal purification. The final product was dispersed in dimethyl sulfoxide (DMSO). Second, PheoA (11.8 mg) was dissolved in 11.8 mL DMSO and activated with EDC (9.4 mg) in the presence of NHS (6.9 mg). After 12 h, the MNP-cystamine solution was added to the activated PheoA solution. The reaction mixture was stirred for 12 h and then dialyzed (MWCO: 1 kDa) against DMSO for 3 days. The obtained PheoA-MNPs were analyzed using a fluorescence spectrometer (Infinite^®^ M200 PRO, TECAN Ltd., Zürich, Switzerland).

### 2.4. Loading of Aromatic Cancer Drugs onto MNPs via π–π Stacking

In this study, Doxo and SN38, which possess tetra- and penta-cyclic rings in their molecular structures, were selected and loaded on the surface of MNPs. A Doxo aqueous solution was mixed with different amounts of PheoA-MNPs to produce Doxo-loaded PheoA-MNPs (Doxo/PheoA-MNPs). After stirring for 12 h, the reaction mixture was centrifuged at 15,000 rpm for 10 min and washed three times with deionized water. The amount of Doxo loaded onto the PheoA-MNPs was determined by ultraviolet-visible (UV-Vis) spectrometry (Cary 100^®^, Agilent Technologies, Santa Clara, CA, USA) at 490 nm.

For SN38, varied concentrations of SN38 were dissolved in DMSO and mixed with the MNPs aqueous solution (water:DMSO, 10:1, *v*/*v*) to produce SN38-loaded MNPs (SN38/MNPs). After stirring for 12 h, the unloaded SN38 was removed by centrifugation at 3000 rpm for 5 min. The amount of loaded SN38 was determined by UV-Vis spectrometry at 390 nm.

### 2.5. Physicochemical Characterization of MNPs

The morphology of the MNPs was visualized by high resolution scanning electron microscopy (HR-SEM^®^ SU8010, Hitachi High-Tech Co., Tokyo, Japan) at 15 kV. The surface charge and hydrodynamic size of the PheoA-MNPs were measured using a dynamic light scattering instrument (Zetasizer^®^ Nano ZS 90, Malvern Instruments Limited, Worc., Malvern, UK). Ultraviolet-visible (UV-Vis) absorption spectroscopy of MNPs was conducted using a UV-Vis spectrophotometer. Fluorescence spectra were recorded on a multimode microplate reader at an excitation wavelength of 408 nm. The functional surface chemistry of MNPs was investigated using an FTIR spectrometer (Spectrum Two^®^, Perkin-Elmer Co., Waltham, MA, USA). An X-ray photoelectron spectrometer (K-Alpha^®^, Thermo Fisher Scientific Co., Waltham, MA, USA) was used to study the valance states of manganese (Mn). The concentration of Mn in MNPs was measured by inductively coupled plasma-mass spectroscopy (ELAN6100^®^, Perkin-Elmer Co., MA, USA). The energy dispersive X-ray (EDX) technique was employed for the elemental analysis of nanoparticles.

### 2.6. Evaluation of Cellular Redox System-Activatable Photodynamic Functions of MNPs

The generation of singlet oxygen was monitored by measuring the decrease in fluorescence intensity of 9,10-dimethylanthracene (DMA). PheoA-MNPs (400 μg/mL) were dispersed in DMSO in the absence or presence of 0.4 mM DTT, and then added to a DMA stock solution (final concentration of 10 μM DMA). The solution was irradiated using a 670 nm laser source (infrared laser, Sloc Laser Co., Shanghai, China) at a light intensity of 150 mW/cm^2^ for 10 min. The decrement in fluorescence intensity of DMA was estimated using a fluorescence spectrometer at an excitation wavelength of 360 nm.

The ROS generated by the photosensitizer accumulate in cancer cells and induce cellular apoptosis. Intracellular accumulation of singlet oxygen generated from photodynamic MNPs was determined via DCFH-DA assays. DCFH-DA is deacetylated to DCFH after cellular disposition and is then converted to fluorescein DCF in the presence of ROS. For the study, human colorectal carcinoma cell lines (HCT 116 cells) (1 × 10^5^ cells/well) were seeded into 24-well plates. After 24 h of incubation, PheoA-MNPs (10 μg/mL) were added to each well and incubated for 1 h; then 0.1 mM glutathione reduced ethyl ester (GSH-Oet) was added and incubated for an additional 2 h to induce the reduction of cystamine linkers. Cells were irradiated with 670 nm NIR-light at a power density of 150 mW/cm^2^ for 5 min. Cells were washed three times with HBSS and labeled with 4 μM of DCFH-DA for 10 min. To remove the unreacted DCFH-DA, cells were washed again with HBSS, and imaged under a fluorescence microscope.

### 2.7. Estimation of 808 nm NIR-Responsive Photothermal Functions of MNPs

To test the photothermal functions of MNPs, MNPs (1 mL) in aqueous solution at different concentrations (0–200 μg/mL) were irradiated with an 808 nm NIR laser at a power density of 1.2 W/cm^2^ for 20 min. The temperature was recorded using an infrared thermal camera (9320 P, Infrared Cameras Inc., Beaumont, TX, USA).

We also estimated the photothermal-triggered release of the drug from MNPs. Each MNP solution (1 mL; 200 μg/mL) was irradiated with an 808 nm NIR laser at a power density of 0.5–2 W/cm^2^ for different times. The solution was centrifuged at 15,000 rpm for 10 min. The amount of released drug was determined using a spectrometer (Cary 100^®^, Agilent Technologies, Santa Clara, CA, USA).

### 2.8. Evaluation of Mn^2+^-Based T_1_-Contrast Effects of MNPs

As mentioned above, MNPs were prepared using the KMnO_4_-oxidation method. We estimated the concentration of Mn ions in the particles using inductively coupled plasma-optical emission spectrometry (ICP-OES). T_1_-contrasting effects of MNPs were evaluated based on the Mn concentration (0‒0.6 mM Mn^2+^). The R^1^ relaxivity, defined as 1/T_1_ with units of s−1, of the MNP solutions was measured at room temperature using a BioSpec 9.4T animal MRI system (Bruker Co., Billerica, MA, USA).

### 2.9. In Vitro Photothermal Cellular Uptake Study

The cellular uptake of Doxo/PheoA-MNPs was assessed using a human colon cancer cell line (HCT 116) by confocal laser scanning microscopy. First, HCT 116 cells were seeded on glass coverslips at a density of 3 × 10^5^ cells/well and allowed to attach at 37 °C for 24 h. Then, the culture medium was replaced with fresh medium supplemented with the same Doxo concentration (8 μg/mL) of PheoA-MNPs/Doxo, or free Doxo. After exposure to MNPs, cells were fixed using paraformaldehyde, washed again, and stained with DAPI (4′,6-diamidino-2-phenylindole) for imaging under a confocal laser scanning microscope (CLSM) (510^®^ META, Carl Zeiss AG, Oberkochen, Germany). Using the same method, the photothermal effect on cellular uptake of MNPs and Doxo was evaluated under 808 nm NIR-treatment.

### 2.10. In Vitro Synergistic Photodynamic and Photothermal Cytotoxic Effects of MNPs

Synergistic effects of combined therapeutic modalities were evaluated as follows. HCT 116 cells (1 × 10^4^ cells per well) were seeded into 96-well plates and incubated in DMEM supplemented with 10% FBS at 37 °C in 5% CO_2_ humidified atmosphere. After 24 h of incubation, cells were treated with 40 μg/mL of MNPs, PheoA-MNPs, or Doxo/PheoA-MNPs for another 2 h to allow cellular uptake. The cells were then irradiated with an 808 nm NIR (1.2 W/cm^2^) or 670 nm NIR (150 mW/cm^2^) for 5 min. The relative cell viability was tested by WST assays, using the same method.

### 2.11. In Vivo Photodynamic and Photothermal Tumor Ablation Studies

All studies involving animals were approved by the Institutional Animal Care and Use Committee (IACUC) of Inha University. An in vivo xenograft model was established to evaluate the anti-cancer efficacy of the designed nanoparticles. Mice were intravenously injected with 20 mg/kg of PheoA-MNPs and Doxo/PheoA-MNPs. All groups were irradiated with 670 nm and 808 nm NIR to perform PDT and PTT, respectively. Temperatures at tumor sites (and images) were measured using a thermal camera. All mice were euthanized and tumors from different groups were collected. The harvested sections were embedded in 4% formalin at room temperature for 24 h, followed by fixation in paraffin blocks. Slices with a thickness of 4 μm were obtained and mounted onto glass slides, and then stained with haemotoxylin and eosin (H&E). The stained areas were examined under an optical microscope. For terminal deoxynucleotidyl transferase deoxyuridine triphosphate nick-end labeling (TUNEL) staining, tumor sections were prepared as described above and incubated with proteinase K, TUNEL reaction mixture (TaKaRaA Bio Inc., Shiga, Japan) and Hoechst 33342 (Invitrogen, Carlsbad, CA, USA).

## 3. Results and Discussion

### 3.1. Characterization of L-DOPA-Derived and COOH End-Capped MNPs

Melanin is a natural biopolymer found in living organisms. The pathway of melanin biosynthesis involves the enzymatic oxidation of dopamine (3,4-dihydroxyphenethylamine) or tyrosine to dopachrome, followed by alteration to 5,6-dihydroxyindole (DHI) or 5,6-dihydroxy-indole-2-carboxylic acid (DHICA), which proceeds further through oxidative polymerization to melanin [15]. Synthetic melanin-like nanoparticles are generally obtained by chemical or enzymatic oxidation of dopamine (3,4-dihydroxyphenethylamine) or L-DOPA (L-3,4-dihydroxyphenylalanine) [16,17,18].

In our study, MNPs were synthesized by KMnO_4_-induced oxidation and self-polymerization of L-DOPA (Appendix A). KMnO_4_ acted as both an oxidizing agent and as an Mn^2+^ donor, which was simultaneously chelated with catechol and/or the carboxylic groups of L-DOPA. For designing purposes of PDT, MNPs were modified with a photosensitizer, pheophorbide a (PheoA), via reducible disulfide linkers for a glutathione-based cellular redox-system responsive dequenching of PheoA. To prepare the PheoA-MNPs, cystamine was introduced to the carboxyl acid group of MNPs through the EDC/NHS crosslinking method. MNPs-cystamine was then coupled with an EDC/NHS activated pheophorbide a to obtain PheoA-MNPs. SEM images showed that both MNPs and PheoA-MNPs were spherical in shape, with a uniform diameter of 140 ± 10 nm (Figure 1A,B). The hydrodynamic sizes of MNPs, PheoA-MNPs, and Doxo/PheoA-MNPs were found to be approximately 110 to 120 nm, as determined by dynamic light scattering (DLS) measurements (Figure 1C). Doxo/PheoA-MNPs presented an increased and right-tailed particle size distribution. The surface Zeta potentials for MNPs, PheoA-MNPs, and Doxo/PheoA-MNPs were found to be −48 mV, +10 mV, and +10 mV, respectively (Figure 1D). The negative surface potential indicated the excessive number of carboxylic acid residues present on MNPs [19]. After the conjugation of MNPs with positively charged cystamine and Doxo, the surface Zeta potential was increased to +10 mV.

Chemical analysis of MNPs was performed by FTIR analysis (Figure 1E). The asymmetric and symmetric peaks representing -CH_2_ stretching of the alkyl chain at 2860 cm^−1^ and 2920 cm^−1^, respectively, indicated successful cystamine dihydrochloride modification of the surface of the MNPs. The primary -NH_2_ bond of cystamine dihydrochloride represented by the peak centered at 3200 cm^−1^ (3200‒3400 cm^−1^ area) disappeared due to the conversion of primary amine to secondary amine, indicating successful conjugation of PheoA with the terminal amine of MNP-cystamine. The surface modification of MNPs with PheoA was further confirmed by analyzing the fluorescence emission spectra (Figure 1F). Fluorescence peaks for both PheoA-MNPs and free PheoA were observed at 670 nm, indicating that the conjugation of PheoA and MNPs did not result in any difference in the fluorescent properties of PheoA. The fluorescence intensity of PheoA-MNPs was found to be lower than that of free PheoA; this was attributed to the quenching effect of MNPs.

### 3.2. Doxorubicin and SN38 Loading on MNPs via π- π Stacking

The chemotherapy drug, Doxo, was loaded onto PheoA-MNPs at various concentrations (0.1–0.8%, (w:w)) to produce Doxo/PheoA-MNPs. After removing the unreacted drug, the drug loading capacity of PheoA-MNPs was determined using UV-Vis spectrometry. The characteristic peak at 490 nm for Doxo was observed in the UV-Vis spectra, indicating the successful loading of Doxo onto PheoA-MNPs (Figure 2B). The maximum drug loading capacity was found to be 35% (Doxo:MNPs, w:w), consistent with that found in the previously reported literature [20,21,22,23]. For instance, Du et al. developed polydopamine nanoparticles conjugated with 33 wt% of Doxo to be used as nanotheranostics for multimodal imaging-guided cancer therapy [20,24]. In another study by Wang et al., Doxo (33 wt%) was loaded onto polyethylene glycol (PEG)-modified polydopamine nanoparticles for chemo-photothermal combination therapy of cancer [25].

Another cancer drug, SN38, whose structure contains six-membered lactone rings, was also introduced into MNPs for drug loading tests (Appendix A). The successful loading of SN38 on FA-MNPs was confirmed based on the existence of two characteristic peaks at 368 nm and 390 nm for SN38 in the UV-Vis spectra (Appendix A).

It was previously shown that melanin-like nanoparticles have the ability to bind to drugs with aromatic structures via hydrogen bonding or π-π stacking [21,25,26,27,28,29,30,31]. In this study, it was also shown that chemotherapeutic drugs with tetra- and pentacyclic rings, can be successfully loaded onto the surface of MNPs. The drug content for the following cytotoxicity and animal studies was set to 10% (*w*/*w*) and 5%(*w*/*w*) for Doxo and PheoA, respectively.

### 3.3. NIR-Responsive Photothermal Effects of MNPs

#### 3.3.1. In Vitro and in Vivo Photothermal Effects of MNPs

To determine their photothermal effects, MNPs at varying concentrations (0‒200 μg/mL) were irradiated using an 808 nm NIR laser at a power density of 1.2 W/cm^2^ for 20 min. It was shown that the temperature of the solutions increases with greater irradiation times and higher concentrations of MNPs (Figure 2D,E). The temperature of the solution containing 200 µg/mL MNPs increased up to around 72 °C while that of water was elevated from 20 °C up to around 25‒30 °C. This considerable increase in temperature demonstrates the effect of strong NIR absorption by MNPs. Repeatable photothermal activity of MNPs was also investigated via sequential irradiation with 10 min cooling-down intervals (Figure 2F). Between each cycle, no temperature decrements or any differences in SEM images were observed, suggesting good photothermal stability of MNPs (Appendix A). These results suggested that MNPs can be utilized in PTT several times for enhanced and reliable therapeutic regimens.

To further evaluate the in vivo performance of MNPs in PTT, whole-body temperature distributions were investigated in tumor-bearing mice. The local temperature of MNPs injected in the tumor increased to 65 °C after 808 nm NIR laser irradiation, leading to tumor ablation (Figure 2G,H). Furthermore, temperatures were not altered in other parts of the mice, which indicated stable distribution and accumulation of MNPs in tumor tissues.

#### 3.3.2. Photothermal Triggered Drug Release

Figure 2C shows the photothermal effects on the release of the drug from Doxo/PheoA-MNPs. The cumulative release of Doxo was found to be nearly 23% after 30 min of NIR irradiation, which is significantly higher than that without NIR irradiation (~5%). The burst release of Doxo triggered by NIR irradiation may be attributed to the photothermal effects of PheoA-MNPs. The increasing temperature may lead to the disassociation of intermolecular interactions between Doxo and MNPs. A similar trend was also observed for the release of SN38 from FA-MNPs (Appendix A).

#### 3.3.3. Calculation of Photothermal Conversion Efficiency of MNPs

A quantitative evaluation of the photothermal conversion efficiency (η) of MNPs was determined using a previously reported equation, as follows [32]:(1)$$\eta =\;\frac{\left(h.A.∆T_{\max}\right)-Q_{0}}{I.\left(1-10^{-A_{\lambda }}\right)}$$

In Equation (1), *h* is the heat transfer coefficient, *A* is the heat transfer area, Δ*T_max_* is the temperature change in the MNP solution, *I* is the laser power, *A_λ_* is the absorbance of MNPs at 808 nm, and *Q*_0_ is the heat dissipated from light absorbed by the solvent. Photothermal efficiency was calculated as 87.65% using Equation (1) for data obtained with 200 μg/mL of MNP solution. The η value of MNPs was remarkably higher than that of other materials such as Au nanorods (20%) [33], PVP Bi nanodots (30%) [34], Cu2-xSe (22%) [33], and polydopamine nanoparticles (40%) [14]. The higher η value, which might be related to numerous π-π interactions and the stacked structure of MNPs, make them a promising PTT agent.

### 3.4. Redox-Responsive Dequenching of PheoA

Experimentally, the self-quenching effect of MNPs was evaluated by examining the fluorescence emission spectra of PheoA-MNPs, MNPs, and free PheoA in DMSO. To understand the reduction-responsive dequenching behavior of MNPs, fluorescence emission spectra were monitored in the presence of dithiothreitol (DTT), which dissociates disulfide bonds. The fluorescence intensity of PheoA-MNPs was remarkably increased after incubation with DTT or GSH (Figure 3B,C). These results demonstrated that the photoactivity of PheoA could be recovered under reductive conditions, with the rapid cleavage of disulfide bonds by DTT and GSH.

The efficiency of Förster resonance energy transfer (FRET) depends mainly on the following: (i) the overlap between the fluorescence emission spectrum of the donor molecule and absorption/excitation spectrum of the acceptor, (ii) the distance between the donor and acceptor (typically 1 to 10 nm), (iii) the parallel orientation of the donor emission dipole moment and the acceptor absorption dipole moment, and (iv) the fluorescence lifetime of the donor molecule [35].

Our data demonstrated that MNPs act as Förster resonance energy transfer (FRET)-based quenchers (acceptors) because of their broad energy absorption bandwidth in the visible range which overlaps with the fluorescence emission spectrum of PheoA (donor) [35,36,37]. Therefore, the photodynamic design of MNPs inevitably requires a specific type of dequenching system, such as a cancer redox-responsive cleavable system.

### 3.5. Cellular Redox System-Activatable Photodynamic ROS Generation by MNPs

To evaluate singlet oxygen generation by PheoA-MNPs, singlet oxygen quantum yields in the absence or presence of DTT were monitored using DMA as a singlet oxygen trap (Figure 3D). PheoA-MNPs, in the absence of DTT, did not cause a noticeable decline in DMA fluorescence intensity upon exposure to the 670 nm NIR laser. However, PheoA-MNPs in the presence of DTT exhibited an obvious decline in DMA fluorescence intensity because of efficient singlet oxygen generation via the dequenching process. Furthermore, GSH-mediated photodynamic activity of PheoA-MNPs against cancer cells was evaluated (Figure 3E). GSH-responsive nano-sized drug delivery systems have been reported for targeted intracellular delivery [38] and photodynamic therapy [39]. These approaches are designed to provide the triggered release of therapeutic agents after entering cells. It is known that intracellular compartments such as cytosol and cell nuclei have significantly higher GSH concentrations (1‒10 mM) than the extracellular environment (2‒20 μM) [39,40]. Furthermore, some cancer cells express higher levels of GSH than normal cells [39,41,42].

### 3.6. T_1_-Weighted MR Imaging Properties of MNPs

MR imaging is a commonly used imaging modality in clinical applications, and is based on the interaction of protons with the surrounding environment. Gadolinium (Gd^3+^) [14,22,43], iron (Fe^3+^) [44,45,46], and manganese (Mn^2+^) [47,48,49,50] are the currently used contrast agents which shorten the T_1_ relaxation time and yield brighter images in T_1_-weighted MRI [51,52]. However, the addition of a contrast agent to a nano-carrier remains a challenging task due to increments in the cost and complexity of synthesis and purification. Fan et al. showed the potential usage of melanin nanoparticles for positron emission tomography and MRI by chelating with Cu^2+^ and Fe^3+^ after synthesizing melanin nanoparticles [53]. Here, we employed a one-pot self-polymerization method to fabricate Mn^2+^-embedded MNPs.

The valence states of Mn^2+^ in MNPs were investigated by X-ray photoelectron spectroscopy (XPS). An Mn 2p narrow scan of MNPs revealed two peaks, at 643 eV and 653 eV, which correspond to Mn 2p_3/2_ and Mn 2p_1/2_, respectively (Figure 4D). Additionally, the presence of MnO was confirmed based on the characteristic satellite feature at 647 eV, which is not present for either Mn_2_O_3_ or MnO_2_. These results showed that MNPs possess a large Mn^2+^ component after the self-polymerization step, without any need for additional modification. For further investigation of the superior Mn^2+^ loading ability of the designed nanoparticles, energy dispersive X-ray analysis was performed for both MNPs utilized in this study and PDA NPs synthesized by auto-oxidation after neutralization of dopamine hydrochloride with NaOH (Appendix A). Elemental analysis showed that MNPs synthesized by oxidation of L-DOPA with KMnO_4_ had 28.16% (wt) manganese, whereas PDA NPs synthesized using NaOH had only 0.08% (wt) (Table 1). These results confirmed that the methodology employed in this study provided a greater amount of Mn^2+^ for embedding into MNPs during polymerization when compared with preparation methods using NaOH, followed by auto-oxidation of dopamine (Figure 4E).

To evaluate the potential usage of MNPs as endogenous contrast agents for T_1_-weighted MRI, MNPs at different concentrations (0‒0.5 mM) were scanned using an animal MRI system. MNPs displayed a concentration-dependent brightening effect (Figure 4B), and the R^1^ relaxivity had an enhanced value of 17.88 mM^−1^s^−1^ (Figure 4C) when compared with other studies employing melanin-like nanoparticles (Liu et al. (Gd^3+^, 6.9 mM^−1^s^−1^ at 1.5 T) [14], Chen et al. (Fe^3+^, 7.524 mM^−1^s^−1^ at 1.5 T) [46], Fan et al. (Fe^3+^, 1.2 mM^−1^s^−1^ at 1.0 T) [53]). Similarly, Sun et al. developed PEG-modified melanin nanoparticles, followed by chelation with Mn^2+^ for dual modal imaging-guided PTT [54]. The R^1^ relaxivity was reported as 18.86 mM^−1^s^−1^ at 3.0 T. In another study, Dong et al., prepared indocyanine green loaded PEG modified polydopamine nanoparticles, which were then loaded with Doxo and chelated with Mn^2+^ ions, for imaging-guided chemo-photothermal combination therapy of cancer [55]. The R^1^ relaxivity of these nanoparticles was found to be 14.15 mM^−1^s^−1^ at 3.0 T. Considering the value of R^1^ based on manganese contrast agents generally increases with the decrease of magnetic field strength, R^1^ relaxivity of 17.88 mM^−1^s^−1^ obtained under 1.5 T might be lower than other studies. However, when taken together with the one-pot preparation strategy, MNPs developed in this study offer opportunities to construct multifunctional nanoplatforms for improved cancer therapy.

### 3.7. Enhanced Cellular Uptake of MNPs via Photothermal Effects

The uptake of free Doxo and Doxo/PheoA-MNPs in HCT 116 cells was evaluated by CLSM. The presence of the red fluorescence signal confirmed the internalization of free Doxo or Doxo/PheoA-MNPs in HCT 116 cells (Figure 5A). NIR laser irradiation did not result in any significant effect on the intensity of fluorescence signals from cells treated with free Doxo. However, the intensity of signals from cells treated with Doxo/PheoA-MNPs was almost 1.7 times higher than that observed without laser exposure (Appendix A). As MNPs work as photothermal agents, this result is consistent with other studies in which PTT was shown to enhance the cellular uptake of drugs or nanoparticles [56,57,58].

To further examine the role of FA in cellular uptake of FA-MNPs, HEY-T30 cells were incubated with MNPs (Appendix A) and FA-MNPs (Appendix A) and cellular uptake was visualized using TEM. Consistent with our hypothesis, increased cellular endocytosis of FA-MNPs (Appendix A) was observed, whereas cells incubated with MNPs showed acceptable cellular uptake due to enhanced permeability and retention (EPR) effects (Appendix A). These results demonstrated the huge potential of the nanoparticles utilized in this study, including targeting and PTT agents.

### 3.8. PDT/PTT Synergistic Cytotoxic Effects of Doxo/PheoA-MNPs

In vitro cytotoxicity of MNPs was determined in HCT 116 and HEY-T30 cells using water-soluble tetrazolium assays. Human colon cancer cells (HCT 116) were incubated with varying concentrations (0‒100 μg/mL) of MNPs, PheoA-MNPs, and Doxo/PheoA-MNPs for 24 h. MNPs and PheoA-MNPs did not exhibit any noticeable cytotoxicity, indicating the cytocompatibility of MNPs as nano-carriers. However, the viability of cells treated with Doxo/PheoA-MNPs was significantly decreased due to chemotherapeutic effects of the drug-loaded nanoparticles (Figure 5B).

The synergistic cytotoxic effects were achieved by combining MNPs, PheoA, and Doxo. In vitro experiments were performed to confirm the anti-tumor effects of Doxo/PheoA-MNPs. Nanoparticle cytotoxicity was studied using WST assays. MNPs and PheoA-MNPs did not exhibit any cytotoxicity to HCT 116 cells in the dark, while the viability of cells treated with Doxo/PheoA-MNPs was remarkably lower, indicating effective chemotherapy (Figure 5C). When cells were irradiated with the 670 nm NIR laser for PDT, PheoA-MNPs showed significantly higher cytotoxicity than MNPs due to the existence of PheoA and the generation of singlet oxygen from PheoA-MNPs, consistent with the results from GSH-mediated photoactivity. The potential of nanoparticles for PTT was evaluated in response to 808 nm NIR irradiation, and cytotoxicity was observed in all groups; the results demonstrated outstanding photothermal behavior. For PTT and PDT, Doxo/PheoA-MNPs showed improved cytotoxicity due to the chemotherapeutic effects of loaded Doxo (Figure 5C).

### 3.9. In Vivo PDT/PTT Anti-Cancer Efficacy of MNPs

The in vivo anti-cancer efficacy of Doxo/PheoA-MNPs was evaluated using H&E and TUNEL staining of tumors in tumor-bearing mice (Figure 6). Histological images of the control group animals revealed normal morphology and nuclear structures without any necrosis in tumor cells (Figure 6A). PheoA-MNPs in response to 670 nm NIR irradiation resulted in apoptosis via PDT effects (Figure 6B). However, Doxo/PheoA-MNPs exhibited more extensive damage, and significant apoptosis in response to 808 nm NIR irradiation by virtue of the combined chemotherapeutic effects of Doxo (Figure 6C).

## 4. Conclusions

Here, we reported a melanin-like poly(L-DOPA) nanoparticle-based drug delivery system with multifunctional properties, including (i) T_1_-weighted imaging capability, (ii) GSH/NIR dual responsive intracellular drug release, and (iii) specific tumor targeting with FA, for combination therapy regimen of PDT, PTT, and chemotherapy. Although there are other reported studies in which melanin-like nanoparticles were used for bimodal therapy [25,57] and monotherapy [14,21,26,59,60], to our knowledge, this is the only study that has used melanin-like poly(L-DOPA) nanoparticles, with a trimodal synergistic therapeutic approach for imaging-guided cancer treatment. Compared with traditional therapeutic strategies, Doxo/PheoA-MNPs are expected to have a great potential in cancer therapy. In addition, they can be used as model nanotherapeutic carriers for the treatment of different cancer types, by virtue of the ability of melanin to bind to many widely used chemotherapeutic drugs with aromatic structures.

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
