# Peer review of "T1-Positive Mn2+-Doped Multi-Stimuli Responsive poly(L-DOPA) Nanoparticles for Photothermal and Photodynamic Combination Cancer Therapy"

_biomedicines, 2020, doi:10.3390/biomedicines8100417_

Round 1
Reviewer 1 Report
The manuscript by Kang et al. reports the development and evaluation of multifunctional nanoparticles for cancer therapeutic and diagnostic applications. The reported nanoparticle system offers many interesting features that are potentially beneficial for the cancer nanomedicine field. I find drug loading mechanism via the aromatic pi-pi interaction very clever. This may be advantageous for the drug release at targeted tissue sites. Drug loading contents in this system is quite and high and the nanoparticles are also very uniform in size. Phototherapy efficacy of this nanoparticle drug delivery system is also great. I am not sure how much MRI imaging capability adds to the value of this drug delivery design for clinical translation prospects, but it is a bonus feature especially given the superior safety profile of Mn over Gd.
The manuscript is well written. Conclusions are well supported by the results. I have a few editorial comments below.
- The title is very long and somewhat confusing. Please consider shortening it.
- Please consider replacing “T1-contrasting” in the title. T1-positive?
- In the abstract, the phrase “based on design objectives” (line 27) does not really add anything to the statement.
- Please make sure that experimental values from similar measurements are reported with the same level of accuracy and precision. For example, temperature values are reported as 71.98 ºC vs 25-30 °C (line 285). Can the temperature measurement from this technique read to two decimal points?
- Section 4. Discussion contains template statements and should be removed. Perhaps, Section 3 can be titled “Results and Discussion” given the authors discussed their results here.
Author Response
Dear Editor and Reviewer
We appreciate the helpful comments from reviewers. Reviewers’ comments are very helpful and appropriate to improve the science of our manuscript. We put our best effort to respond their suggestions and carefully revised the manuscript according to the reviewers’ comments. We sincerely hope the amended data and manuscript will be satisfactory for the peer-review publication.
Best,
Sugeun

Reviewer 2 Report
The paper reports an interesting study of multi-functional nanoplatform for cancer therapy using melanin-like poly(L-DOPA) nanoparticles. The experimental data are largely sound. Thus, I recommend the publication of this work after addressing several minor issues listed below:
1) Page 13, line 436, when the Doxo-loaded PheoA-MNPs concentration ranges from 1 to 100 ug/mL, what are the equivalent concentrations of PheoA and Doxo?
2) Figure 5A, what are the black spots in overlay image of Dox/PhoeA-MNP alone? Authors need to provide colloidal stability data of Dox/PhoeA-MNP in biological media.
3) Page 5, line 207, authors used only one dose (20 mg/kg). Is there a reason for using that concentration?
4) Page 14, the discussion on the Results is missing.
Author Response

(The authors gave the same response as above.)
